# Comparison of the Metabolic Profile between Unstimulated and Stimulated Saliva Samples from Pregnant Women with/without Obesity and Periodontitis

**DOI:** 10.3390/jpm13071123

**Published:** 2023-07-11

**Authors:** Gerson Aparecido Foratori-Junior, Adrien Le Guennec, Tatiana Kelly da Silva Fidalgo, James Jarvis, Victor Mosquim, Marília Afonso Rabelo Buzalaf, Guy Howard Carpenter, Silvia Helena de Carvalho Sales-Peres

**Affiliations:** 1Department of Pediatric Dentistry, Orthodontics and Public Health, Bauru School of Dentistry, University of São Paulo, Bauru 17012-901, Brazil; 2Nuclear Magnetic Resonance Facility, Guy’s Campus, King’s College London, London SE1 1UL, UK; 3Department of Preventive and Community Dentistry, School of Dentistry, Rio de Janeiro State University, Rio de Janeiro 20551-030, Brazil; 4Randall Division of Cell and Molecular Biophysics and Centre for Biomolecular Spectroscopy, Guy’s Campus, King’s College London, London SE1 1UL, UK; 5Department of Operative Dentistry, Endodontics and Dental Materials, Bauru School of Dentistry, University of São Paulo, Bauru 17012-901, Brazil; 6Department of Biological Sciences, Bauru School of Dentistry, University of São Paulo, Bauru 17012-901, Brazil; 7Centre for Host-Microbiome Interactions, Faculty of Dental, Oral & Craniofacial Sciences, Guy’s Campus, King’s College London, London SE1 1UL, UK

**Keywords:** metabolomics, obesity, periodontitis, pregnancy, saliva

## Abstract

This study aimed to compare the metabolic profile of unstimulated (US) and stimulated (SS) saliva samples from pregnant women with/without obesity and periodontitis. Ninety-six pregnant women were divided into: obesity + periodontitis (OP = 20); obesity/no periodontitis (OWP = 27); normal BMI + periodontitis (NP = 20); and normal BMI/no periodontitis (NWP = 29). US and SS samples were collected by expectoration and chewing of sterilized parafilm gum, respectively, and samples were individually analyzed by Proton Nuclear Magnetic Resonance (^1^H-NMR). Univariate (*t* test and correlations) and multivariate (Principal Component Analysis–PCA, and Partial Least Square–Discriminant Analysis—PLS-DA with Variance Importance Projection–VIP scores) and Metabolite Set Enrichment Analysis were done (*p* < 0.05). Metabolites commonly found in all groups in elevated concentration in US samples were 5-Aminopentoate, Acetic acid, Butyric acid, Propionic acid, Pyruvic acid, and Succinic acid. They were mainly related to the butyrate metabolism, citric acid cycle, amino sugar metabolism, fatty acids biosynthesis, pyruvate metabolism, glutamate metabolism, and Warburg effect. Metabolites commonly found in all groups that were in elevated concentration in SS samples were Citrulline, Fumaric acid, Histidine, *N*-acetyl glutamine, *N*-acetylneuraminic acid, para-hydroxyphenylacetic acid, Proline, Tyrosine. Although some differences were found between unstimulated and stimulated saliva samples from pregnant women with/without obesity and periodontitis, stimulated saliva collection seems adequate, demonstrating similar metabolic pathways to unstimulated saliva samples when groups are compared.

## 1. Introduction

Several studies have investigated the epidemiological association between obesity and periodontitis in pregnant women [1]. Nonetheless, the pathophysiological understanding of the association of these outcomes is not clear. Salivary analysis has been widely used to understand the individual’s biological mechanisms related to health or diseases. In addition to being an easily accessible and non-invasive method, saliva has a variety of proteins and metabolites that can be studied as potential biomarkers of systemic and oral diseases [2]. In some situations, collecting unstimulated saliva from individuals may be challenging due to systemic or oral changes that impair salivary flow, as occurs in obesity, pregnancy, and periodontitis. Then, it is essential to methodologically assess whether there are differences in the components in unstimulated and stimulated saliva samples. Therefore, this study aimed to compare the metabolic profile of unstimulated and stimulated saliva samples from pregnant women with/without obesity and periodontitis by Proton Nuclear Magnetic Resonance (^1^H-NMR).

Periodontitis is a multifactorial chronic inflammatory disease that leads to loss of periodontal attachment on the tooth root surface and resorption of alveolar bone [3]. It is a consensus that periodontitis is dependent on dental biofilm dysbiosis [4], but as there is an increase in the number of systemic diseases and conditions that are linked to periodontitis, the extent and severity of tissue destruction appears to be influenced by host characteristics [5].

Periodontal medicine refers to understanding the interrelationship between periodontal diseases and systemic changes in the body. This field seeks to integrate periodontal knowledge into the holistic care of patients, aiming at comprehensive care and quality of life for individuals. The number of diseases and conditions linked to periodontitis has increased exponentially in recent years [6]. A 2016 systematic review of clinical trials on periodontal medicine indicated 57 conditions potentially related to periodontal diseases [5,6].

The relationship between periodontitis and pregnancy in women with obesity has been currently investigated [1]. Periodontal diseases and pregnancy have been extensively studied. Their associations can be explained by the reduction of the antimicrobial activity of peripheral neutrophils, the increase in the proportion of aerobic and anaerobic bacteria (*Bacteroides melaninogenicus*, *Prevotella intermedia* and *Porphyromonas gingivalis*), and the changes in the periodontal connective tissue turnover and the high levels of estrogen and progesterone [7]. Similarly, obesity may trigger a generalized inflammatory response in the body of individuals due to the production of pro-inflammatory cytokines, adipokines, and other bioactive substances by adipose tissue [8,9]. Also, the immune response of the individual with obesity is impaired due to changes in T lymphocytes and monocytes/macrophages. Faced with this, individuals with obesity become more prone to inflammation in the oral cavity, even in the presence of a small amount of biofilm [9,10]. It is hypothesized that these biological interactions during pregnancy become more intense when associated with immunological damage from overweight/obesity [1].

With technological advances, many studies sought to identify disease biomarkers through body fluids. Saliva is one of the fluids extensively studied due to its easy collection method. Previously, salivary proteomic [11,12,13,14] and metabolomic [15,16,17,18,19,20,21] analyses related to periodontitis were made. Also, although there is a scarcity of studies that evaluated the salivary metabolic profile during pregnancy, salivary proteomic [22,23,24,25] analyses were already performed during pregnancy before. Specifically, our group recently investigated the proteomic and metabolomic profiles of unstimulated saliva from pregnant women with obesity and periodontitis [26,27]. The proteomic analysis highlighted several differentially expressed proteins, and potential biomarkers, associated with obesity and periodontitis separately.

Nonetheless, we called attention to the importance of those up- or down-regulated proteins when obesity and periodontitis were present in combination during pregnancy, such as *Submaxillary gland androgen-regulated protein 3B*, *Protein S100-A8*, *Matrix metalloproteinase-9* (MMP9), *Heat shock 70 kDa protein 2* and *6*, *Putative Heat shock 70 kDa protein 7*, *Heat shock 71 kDa protein*, *Haptoglobin*, *Plastin-1*, *Prolactin-inducible protein*, and *Alpha-defensins 1* and *3* [26]. The metabolomic analysis, in turn, demonstrated a larger difference in metabolic activity in unstimulated saliva samples mainly related to periodontitis, regardless of obesity. Periodontal parameters were positively correlated with many saliva metabolites (i.e., leucine, ornithine, isovaleric acid, valine, isoleucine, putrescine, taurine, phenylalanine, propionic acid, and acetic acid) [27]. The main metabolic pathways found in unstimulated saliva samples from pregnant women with/without obesity and periodontitis were the glucose-alanine cycle; alanine metabolism; valine, leucine and isoleucine degradation; glutamate metabolism; and Warburg effect, indicating amino acids; saturated fatty acids; and straight chain fatty acids as the main subclasses of metabolites [27].

Nonetheless, it is essential to point out that one of the challenges in the abovementioned studies was the difficulty in collecting saliva from pregnant women due to frequent nausea and, mainly, the reduced flow in some patients. So, an alternative to this would be the collection of stimulated saliva samples through the tasteless parafilm gum, resulting in a greater amount of saliva samples and facilitating clinical practice. For that, as aforementioned, it is essential to methodologically compare unstimulated and stimulated saliva samples to identify if they are similar or if there are essential changes in their components that could invalidate the method based on the analysis of stimulated saliva.

Our group also compared the proteomic profile between unstimulated and stimulated saliva samples from pregnant women with/without obesity and periodontitis. Our results indicated that saliva stimulation decreased essential proteins involved with immune response and inflammation process in all groups (i.e., *Antileukoproteinase*, *Lysozyme C*, *Alpha-2-macroglobulin-like protein 1*, *Heat shock proteins—70 kDa 1-like*, *1A*, *1B*, *6*, *Heat shock-related 70 kDa protein 2*, *Putative Heat shock 70 kDa protein 7*, *Heat shock cognate 71 kDa*). Additionally, proteins related to the carbohydrate metabolic process and glucose metabolic process were absent in stimulated saliva, mainly in those groups with obesity (with/without periodontitis) (i.e., *Frutose-bisphosphate aldose A*, *Glusoce-6-phosphate isomerase*, *Pyruvate kinase*) [28]. However, there are no studies yet evaluating if there are differences in the metabolic profile of stimulated saliva samples compared to the unstimulated saliva samples in that target population.

Therefore, our research question was, “are there differences in the metabolic profile when unstimulated and stimulated saliva samples from pregnant women with/without obesity and periodontitis are compared?”. The null hypothesis adopted in this study was that there are no differences in the metabolic profile of stimulated saliva compared to unstimulated saliva samples from pregnant women with/without obesity and periodontitis. The alternative hypothesis, in turn, was that salivary stimulation significantly altered salivary metabolites related to obesity and/or periodontitis compared to unstimulated saliva, therefore not being the primary method of choice for metabolomic analysis.

## 2. Materials and Methods

This observational, cross-sectional, and analytical study followed the Strengthening the Reporting of Observational Studies in Epidemiology (STROBE) guidelines [29]. This study was developed in accordance with the Declaration of Helsinki and obtained approval from the Internal Ethics Committee from the Bauru School of Dentistry, University of São Paulo (CAAE 06624519.3.0000.5417; protocol code 3.284.822; approved on 17 April 2019).

### 2.1. Sample Recruitment

Inclusion criteria adopted in this study were: pregnant women, during the 3rd trimester of pregnancy, aged 18–40 years old, with regular registration in the public health system in Brazil and with regular follow-up with an obstetrician (at least one prenatal medical visit per trimester). Women should present adequate cognitive function during pregnancy without impairments that require absolute rest. In contrast, the team adopted the following exclusion criteria before the beginning of recruitment: twin pregnancy, individuals who had neuromotor impairments, arterial hypertension during pregnancy (blood pressure 140/90 mmHg), gestational diabetes mellitus (hyperglycemia: 92 mg/dL—fasting level;  180 mg/dL—after 1 h; and  153 mg/dL—after 2 h); malnutrition (BMI < 18.50 kg/m^2^), overweight (BMI between 25.00 kg/m^2^ and 29.99 kg/m^2^), confirmed or suspected diagnosis of SARS-CoV-2 infection at the moment of the first appointment, hyposalivation (<0.25 mL/min flow rate), who were taking antibiotics or had taken antibiotics during pregnancy, who were taking any medication that could interfere with periodontal status and/or salivary flow (e.g., immunosuppressive drugs, anticonvulsants or calcium channel blockers), who were under orthodontic treatment or any dental treatment with another professional, who had multiple tooth loss (more than two teeth per hemiarch), who had stage IV periodontitis, self-reported systemic disease besides obesity, and users of alcohol/tobacco/illicit drugs [27]. These rigorous inclusion and exclusion criteria were adopted to reach the most homogenous sample possible, avoiding biases or confounders in analysing the metabolic profile of unstimulated and stimulated saliva. Therefore, it was expected that by adopting these criteria and having a good sample pairing, we would avoid great variability in the metabolomic analysis and better understand the role of the study outcomes (obesity and periodontitis).

Initially, 126 pregnant women, aged 18–40 years old, during the 3rd trimester of pregnancy (27th–39th gestational week) were consecutively recruited from Primary Healthcare in Bauru, São Paulo, Brazil, between November 2020 and August 2021. After exclusion, ninety-eight pregnant women were divided into: with obesity and periodontitis (OP = 20); with obesity but without periodontitis (OWP = 27); with normal BMI but with periodontitis (NP = 21); and with normal BMI and without periodontitis (NWP = 30). After the NMR analysis of stimulated saliva samples (highlighted in detail below), two individuals were excluded due to poor water suppression of the stimulated saliva samples. Therefore, the final sample was composed of 96 pregnant women divided into: those with obesity and periodontitis (OP = 20); with obesity but without periodontitis (OWP = 27); with normal BMI but with periodontitis (NP = 20); and with normal BMI and without periodontitis (NWP = 29). The sample size was based on previous reports [16,17,18,19,20,27]. All of them were matched by socioeconomic level.

### 2.2. Sample grouping and examiner standardization

To group individuals, pre-pregnancy nutritional status based on the cut-off point of BMI proposed by the World Health Organization (WHO) was considered. Periodontal status was based on measurements of probing pocket depth (PPD) and clinical attachment level (CAL)/attachment loss (AL), and then patients were classified as having or not having periodontitis. All data collection was performed by one dentist previously calibrated by a gold standard examiner in epidemiological surveys to ensure uniformity in data collection regarding the periodontal conditions. The kappa coefficient (kappa inter-examiner reliability = 0.92; kappa intra-examiner reliability = 0.95) was calculated based on a periodontal diagnosis of approximately 10% of the sample (*n* = 10). A 15-day interval was observed between the examinations performed by the principal examiner dentist and the gold standard examiner because of periodontal alteration after the first examination. The examiner’s training occurred through lectures, study and discussion of didactic material, and demonstrative practical activity with non-pregnant individuals. This training took place to obtain fluidity in the service, correct application of the questionnaires, correct access to the system to obtain previous data (pre-pregnancy weight, information from previous prenatal medical consultations etc.), correct periodontal examination to obtain the PPD and CAL/AL parameters, correct collection of saliva samples. The examiner was instructed to supervise the entire saliva collection to prevent the patients from talking during the process or swallowing the saliva produced. The examiner was also trained to supervise the collection of stimulated saliva, guiding the participants to chew the parafilm gum properly, also avoiding the risk of them swallowing the parafilm or providing support if they felt nauseous.

As mentioned, pre-pregnancy BMI was considered according to the WHO classifications and previous studies [26,27,28,30,31]. Pregnant women with pre-pregnancy BMI ≥ 30.00 kg/m^2^ were allocated to the OP group or the OWP group, while those with normal BMI (18.5 kg/m^2^–24.99 kg/m^2^) were allocated to the NP group or the NWP group. One patient with a borderline BMI of 18.3 kg/m^2^ was included in the NWP group. Pre-pregnancy weight was based on medical files from Primary Healthcare, while women’s height was obtained using a stadiometer (Wood 2.20; WCS Ind., Curitiba, Paraná, Brazil). Patients were diagnosed with periodontitis according to the classification described by Tonetti et al. (2018) [32] and were allocated to the OP or NP groups. Afterwards, periodontitis cases were categorized in stages according to the severity of the disease [32]. The diagnosis and severity of the periodontitis were based only on the clinical parameters, and no dental radiographs were taken to avoid unnecessary exposure of pregnant women to X-rays.

### 2.3. Saliva Collection and Preparation for ^1^H-NMR Analysis

Saliva collection and preparation were previously described [27]. In summary, considering the circadian rhythm, saliva samples were collected in the morning (09:00–11:00), at least 1 h after oral exposure to exogenous substances (eating, chewing gum, or oral hygiene) or exercise [27]. Unstimulated whole saliva (US) was collected by expectoration into sterilized falcon tubes (50 mL) and immersed on ice for 10 min. For the stimulated whole saliva (SS) collection, patients were instructed to chew a sterilized parafilm gum for 1 min and to discard the produced amount. Afterwards, patients were instructed to keep chewing the parafilm gum. The total amount of saliva produced in the following 10 min was put into a sterilized plastic falcon tube (50 mL) immersed in ice [28,31]. Saliva volume and flow rate were recorded. Samples were centrifuged at 15,000× *g* for 10 min at 4 °C. The supernatant was collected and stored at −80 °C until the analysis.

Saliva samples (US and SS) were removed from the freezer and underwent only a single thawing cycle (at room temperature) before ^1^H-NMR analysis. For the ^1^H-NMR analysis, a 540 µL aliquot of each saliva sample (both US ad SS) was mixed with 60 µL of NMR phosphate buffer in 5 mm outside diameter (OD) NMR tubes. NMR buffer for saliva was prepared as follows: 3 parts (*V/V*) deuterium oxide (99.9 atom % D, to provide a field frequency lock) (Sigma-Aldrich Corp. Milwaukee, WI, USA), and 1 part (*V/V*) stock of 2 mM of trimethylsilylpropanoic acid (TSP) [27]. The pH was adjusted to 7.4.

### 2.4. ^1^H-NMR Spectroscopy Analysis

Spectral acquisition and processing were described elsewhere [27,33]. In summary, Bruker Avance NEO 600 MHz equipped with a TCI Cryoprobe Prodigy (Bruker Biospin, Karlsruhe, Germany), operating at a proton frequency of 600.2 MHz at 298 K was used for spectral acquisition. ^1^H spectra were acquired using Carr-Purcell-Meiboom-Gill (CPMG) pulse sequence with water suppression by presaturation [34], with 32 scans, 65,536 points for the free induction decay (FID) during the acquisition, a spectra width of 20.8 ppm, an acquisition time of 2.62 s, and an inter-scan delay of 4 s. TSP peak (0 ppm) was used as internal reference/standard.

Spectra were analyzed in TopSpin (version 4.0.3; Bruker Biospin, Karlsruhe, Germany). A 0.3 Hz exponential line broadening function was applied before Fourier transformation and automatic phase correction. Baselines were inspected, and polynomial baseline correction was applied. Then, spectra were exported into MATLAB software (version 2018b; MathWorks, Natick, MA, USA), where post-processing (alignment and normalization) was handled. The water region (4.2–5.1 ppm) and TSP peak were excluded before normalization. Peak alignment by fast Fourier transform (PAFFT method), and normalization by probabilistic quotient normalization (PQN) [35] were performed using a custom toolbox written by Clendinen et al. [36]. Metabolite identification was performed simultaneously based on the human metabolome database (HMDB, http://www.hmdb.ca) (accessed on 19 September 2022). The resulting list of peak areas for each metabolite was converted to a .csv file to be exported to MetaboAnalyst 5.0 software (www.metaboanalyst.ca) (accessed on 20 September 2022), which was used for data analysis. Pareto scaling was also applied on MetaboAnalyst for normalization. 

### 2.5. Statistical Analysis and Bioinformatics 

Data were presented as mean and standard deviation, median and 1st–3rd quartiles, or percentages for clinical parameters. Statistical analysis was performed with IBM SPSS Version 25 (IBM Corp. Released 2017. IBM SPSS Statistics for Windows, Version 25.0. Armonk, NY, USA: IBM Corp.). Quantitative variables were first tested for normality using the Kolmogorov-Smirnov test. One-way ANOVA with Tukey’s test as post-hoc was used for quantitative variables with normal distribution. In contrast, Kruskal-Wallis with Dunn’s test as post-hoc was used for quantitative variables without normal distribution. A significance level of 5% was considered.

For metabolomic analysis, univariate analysis was handled through *t* test and correlations using the MetaboAnalyst 5.0 software (www.metaboanalyst.ca) (accessed on 20 September 2022) to evaluate whether the overall comparison was significant among US and SS for each group. Multivariate analysis was based on Principal Component Analysis (PCA) and Partial Least Square—Discriminant Analysis (PLS-DA) to obtain the predictive performance of the models; each model was evaluated for Q^2^, R^2^, and accuracy (ACC). For each model, 1000 permutations were performed. The variable importance in projection (VIP) scores was obtained based on PLS-DA to determine the relative abundances of the main 15 metabolites that contributed to the separation between US and SS. There are two important measures in PLS-DA: the variable importance in projection (VIP) and the weighted sum of absolute regression coefficients. The colored boxes on the right of each figure of the results presented in this manuscript (see figures in the results section; Figure 1, Figure 2, Figure 3 and Figure 4) indicate the relative concentrations of the corresponding metabolite in each subgroup under study (in this case, US and SS samples). All these multivariate analyses were also handled on MetaboAnalyst 5.0 software (www.metaboanalyst.ca) (accessed on 20 September 2022) after sample normalization by sum, Log transformation (base 10), and data scaling by Pareto scaling (mean-centred and divided by the square root of the standard deviation of each variable).

Metabolite set enrichment analyses (MSEA) were conducted for visualization and functional analysis of metabolites [27,37] on MetaboAnalyst 5.0 software (www.metaboanalyst.ca) (accessed on 20 September 2022). Only metabolites highlighted in the VIP score that also presented *p* < 0.05 in the univariate analysis in common for each group were included in the MSEA for US and SS samples. In the MSEA for US samples, the metabolites included were 5-Aminopentoate, Acetic acid, Butyric acid, Propionic acid, Pyruvic acid, and Succinic acid. In the MSEA for SS samples, the metabolites included were Citrulline, Fumaric acid, Histidine, *N*-acetyl glutamine, *N*-acetylneuraminic acid, para-hydroxyphenylacetic acid, Proline, and Tyrosine.

As a supplement, we performed the metabolomic analysis comparing groups for stimulated saliva samples (see Appendix A). For this, we adopted one-way ANOVA and Tukey test as post-hoc for the univariate analysis using the IBM SPSS Version 25 (IBM Corp. Released 2017. IBM SPSS Statistics for Windows, Version 25.0. Armonk, NY, USA: IBM Corp.) to compare the normalized concentrations of each metabolite (showing the mean concentration and its standard deviation). We also reported the multivariate analysis through the scores plot between the first two principal components (PCs) of sPLS-DA and the loading plot of component 1 from the sPLS-DA. As aforementioned, multivariate analysis was handled on MetaboAnalyst 5.0 software (www.metaboanalyst.ca) (accessed on 5 July 2023) after sample normalization by sum, Log transformation (base 10), and data scaling by Pareto scaling (mean-centred and divided by the square root of the standard deviation of each variable). MSEA were also conducted for visualization and functional analysis of metabolites in the intergroup comparations for stimulated saliva samples. Metabolites that were in higher or lower concentrations according to the univariate and multivariate analyses in the intergroup comparations of SS samples were adequately included in the MSEA, using the MetaboAnalyst 5.0 software (www.metaboanalyst.ca) (accessed on 5 July 2023).

## 3. Results

The mean age of the sample was 27.36 years ± 5.51 (OP: 30.20 ± 5.21; OWP: 27.22 ± 5.99; NP: 25.15 ± 5.12; NWP: 26.89 ± 5.70). Table 1 shows the comparison among groups for anthropometric and periodontal variables, showing the distribution of cases of periodontitis in the different stages of the disease. Mean (± SD) flows of US and SS were, respectively, 0.74 ± 0.06 and 1.69 ± 0.16 for OP; 0.69 ± 0.05, and 1.71 ± 0.14 for OWP; 0.71 ± 0.05, and 1.70 ± 0.14 for NP; and 0.68 ± 0.04, and 1.73 ± 0.12 for NWP.

Appendix A shows the intergroup comparations for SS samples. In summary, the scores plot between the first two principal components (PCs) of sPLS-DA indicates mainly a difference of OP and NP groups from the controls (OWP and NWP), but with no significant distinction between them. According to the univariate analysis, OP and NP groups showed higher concentrations of butyric acid, isovaleric acid, leucine, valine, isoleucine, propionic acid, acetic acid, trimethylamine, ornithine, and phenylalanine. Besides the higher levels of propionic acid, butyric acid, and acetic acid, the OP and NP groups showed lower concentrations of proline, *N*-acetyl glutamine acid, citric acid, pyruvic acid, lactate, *N*-acetylneuraminic acid, and galactose, according to the multivariate analysis by loading factor of component 1 from sPLS-DA. We described the MSEA for visualization and functional analysis of metabolite data. The three top pathways related to the metabolites in higher concentration in OP and NP groups were: Valine, leucine, and isoleucine degradation; Fatty acid biosynthesis; and Propanoate metabolism (Appendix A). The main metabolic subclasses related to the metabolites in higher concentration in OP and NP groups were: Amino acids, Saturated fatty acids, and Straight chain fatty acids (Appendix A). The five top pathways related to the metabolites that were in lower concentration in OP and NP groups were: the Warburg effect; Transfer of acetyl groups into mitochondria; Citric acid cycle, Amino sugar metabolism, and Gluconeogenesis (Appendix A). The main metabolic subclasses related to the metabolites in lower concentration in OP and NP groups were: Amino acids, *N*-acylneuraminic acids, and Short-chain acids and derivatives (Appendix A).

For the comparison between US and SS from the OP group, 47 metabolites were identified, and 13 were significant in the univariate analysis (6 higher in the US and 7 higher in SS). Butyric acid (*p* = 1.31 × 10^−4^), Propionic acid (*p* = 7.81 × 10^−4^), Pyruvic acid (*p* = 8.80 × 10^−3^), 5-Aminopentoate (*p* = 0.0262), Acetic acid (*p* = 0.0291), and Succinic acid (*p* = 0.0324) were in higher concentration in US samples when compared to SS, while para-Hydroxyphenylacetic acid (*p* = 2.84 × 10^−11^), *N*-acetylneuraminic acid (*p* = 5.30 × 10^−7^), histidine (*p* = 1.08 × 10^−5^), fumaric acid (*p* = 6.57 × 10^−4^), proline (*p* = 1.65 × 10^−3^), and Tyrosine (*p* = 0.0198) were in higher concentrations in SS samples. One unidentified metabolite was higher in SS (Figure 1A,B). Multivariate analysis can be found in Figure 1C–E, highlighting the score plot of PCA (C), the score plot between the first two principal components of PLS-DA (D) (ACC = 0.94; R^2^ = 0.89; Q^2^: 0.71), and VIP scores based on PLS-DA (E). VIP scores of the PLS-DA indicated that Acetic acid, Propionic acid, Butyric acid, Succinic acid, Pyruvic acid, 5-Aminopentoate, Maltose, Glucose, 2,3-Butanediol, and Formic acid were in higher concentration in US samples compared to SS, while para-Hydroxyphenylacetic acid, *N*-acetylneuraminic acid, Citrulline, and Histidine were in higher concentration in SS samples. Appendix A shows the concentration differences between US and SS in the OP group for those metabolites highlighted in the VIP score and presented *p* < 0.05 in the univariate analysis.

**Figure 1 jpm-13-01123-f001:**
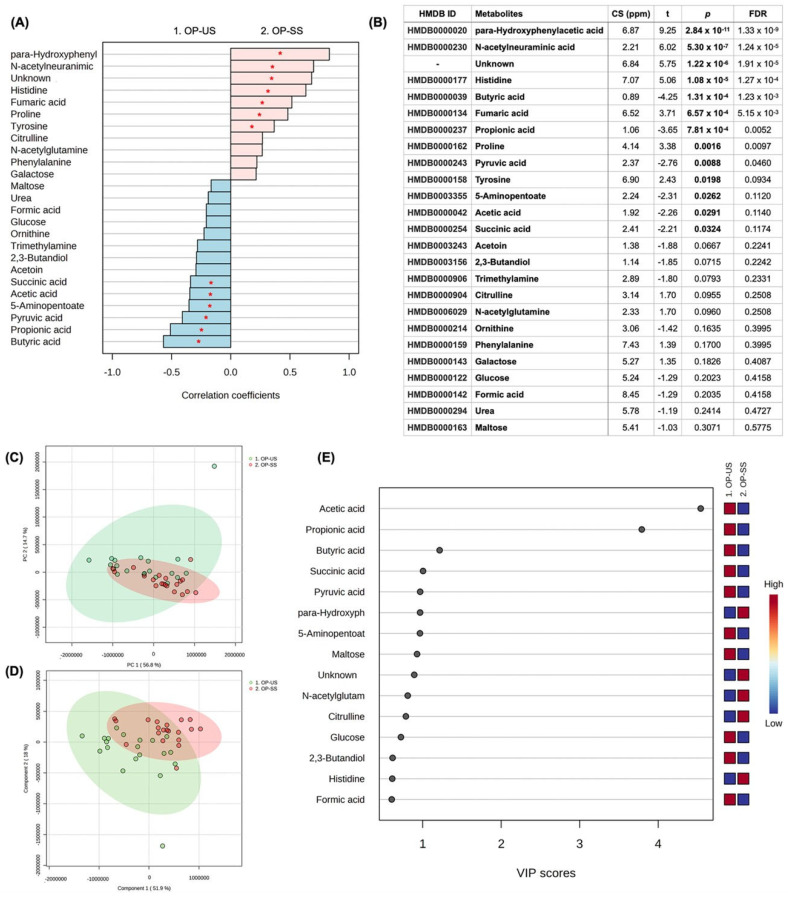
Univariate and multivariate analysis among US and SS samples from the OP group. (**A**) Correlation pattern. Red stars indicate *p* < 0.05. (**B**) Univariate analysis (*t* test); CS (ppm): chemical shift. (**C**) Score plot of PCA. (**D**) Score plot of the 1st and 2nd components of PLS-DA. (**E**) VIP scores of PLS-DA.

For the comparison between US and SS from the OWP group, 47 metabolites were identified, and 19 were significant in the univariate analysis (10 higher in the US and nine higher in SS). Butyric acid (*p* = 1.69 × 10^−5^), Propionic acid (*p* = 3.05 × 10^−5^), 5-Aminopentoate (*p* = 0.0002), Ornithine (*p* = 0.0002), Acetic acid (*p* = 0.0012), Glucose (*p* = 0.0076), Trimethylamine (*p* = 0.0097), Pyruvic acid (*p* = 0.0123), Acetoin (*p* = 0.0130), and 2,3-Butanediol (*p* = 0.0255) were in higher concentration in US samples when compared to SS, while para-Hydroxyphenylacetic acid (*p* = 1.69 × 10^−16^), *N*-acetylneuraminic acid (*p* = 2.21 × 10^−10^), Histidine (*p* = 9.99 × 10^−9^), Proline (*p* = 1.33 × 10^−7^), *N*-acetyl glutamine (*p* = 0.0016), Fumaric acid (*p* = 0.0019), Citrulline (*p* = 0.0021), and Tyrosine (*p* = 0.0072) were in higher concentrations in SS samples. One unidentified metabolite was higher in SS (Figure 2A,B). Multivariate analysis can be found in Figure 2C–E, highlighting the score plot of PCA (C), the score plot between the first two principal components of PLS-DA (D) (ACC = 0.98; R^2^ = 0.93; Q^2^: 0.86), and VIP scores based on PLS-DA (E). VIP scores of the PLS-DA indicated that Acetic acid, Propionic acid, 5-Aminopentoate, Lactate, Succinic acid, Glucose, Butyric acid, Pyruvic acid, Maltose, 2,3-Butanediol, and Ornithine were in higher concentrations in US samples compared to SS, while *N*-acetylneuraminic acid, para-Hydroxyphenylacetic acid, and Histidine were in higher concentration in SS samples. Appendix A shows the concentration differences between US and SS in the OWP group for those metabolites highlighted in the VIP score and presented *p* < 0.05 in the univariate analysis.

**Figure 2 jpm-13-01123-f002:**
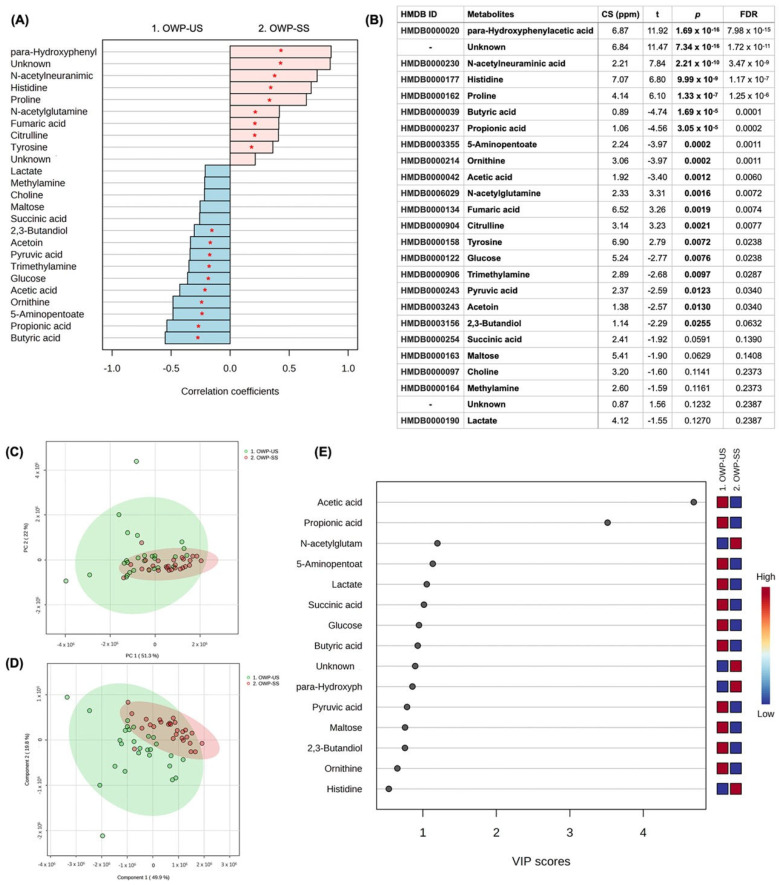
Univariate and multivariate analysis among US and SS samples from the OWP group. (**A**) Correlation pattern. Red stars indicate *p* < 0.05. (**B**) Univariate analysis (*t* test); CS (ppm): chemical shift. (**C**) Score plot of PCA. (**D**) Score plot of the 1st and 2nd components of PLS-DA. (**E**) VIP scores of PLS-DA.

For the comparison between US and SS from the NP group, 47 metabolites were identified, and 22 were significant in the univariate analysis (11 higher in the US and 11 higher in SS). 2,3-Butanediol (*p* = 1.11 × 10^−5^), Butyric acid (*p* = 4.25 × 10^−5^), Pyruvic acid (*p* = 0.0016), Propionic acid (*p* = 0.0018), Acetoin (*p* = 0.0025), 5-Aminopentoate (*p* = 0.0056), Ornithine (*p* = 0.0072), Acetic acid (*p* = 0.0104), Trimethylamine (*p* = 0.0356), Choline (*p* = 0.0380), and Succinic acid (*p* = 0.0486) were in higher concentration in US samples when compared to SS, while para-Hydroxyphenylacetic acid (*p* = 2.64 × 10^−12^), *N*-acetylneuraminic acid (*p* = 9.84 × 10^−9^), Proline (*p* = 1.88 × 10^−5^), Histidine (*p* = 2.52 × 10^−5^), Tyrosine (*p* = 0.0001), Fumaric acid (*p* = 0.0026), Glycine (*p* = 0.0027), Citrulline (*p* = 0.0033), Phenylalanine (*p* = 0.0336), and *N*-acetyl glutamine (*p* = 0.0412) were in higher concentrations in SS samples. One unidentified metabolite was higher in SS (Figure 3A,B). Multivariate analysis can be found in Figure 3C–E, highlighting the score plot of PCA (C), the score plot between the first two principal components of PLS-DA (D) (ACC = 1.00; R^2^ = 0.94; Q^2^: 0.86), and VIP scores based on PLS-DA (E). VIP scores of the PLS-DA indicated that Acetic acid, Propionic acid, Butyric acid, 5-Aminopentoate, Pyruvic acid, Succinic acid, 2,3-Butanediol, Ornithine, and Formic acid were in higher concentrations in US samples compared to SS, while *N*-acetyl glutamine, para-Hydroxyphenylacetic acid, Glycine, Tyrosine, and Citrulline were in higher concentration in SS samples. Appendix A shows the concentration differences between US and SS in the NP group for those metabolites highlighted in the VIP score and presented *p* < 0.05 in the univariate analysis.

**Figure 3 jpm-13-01123-f003:**
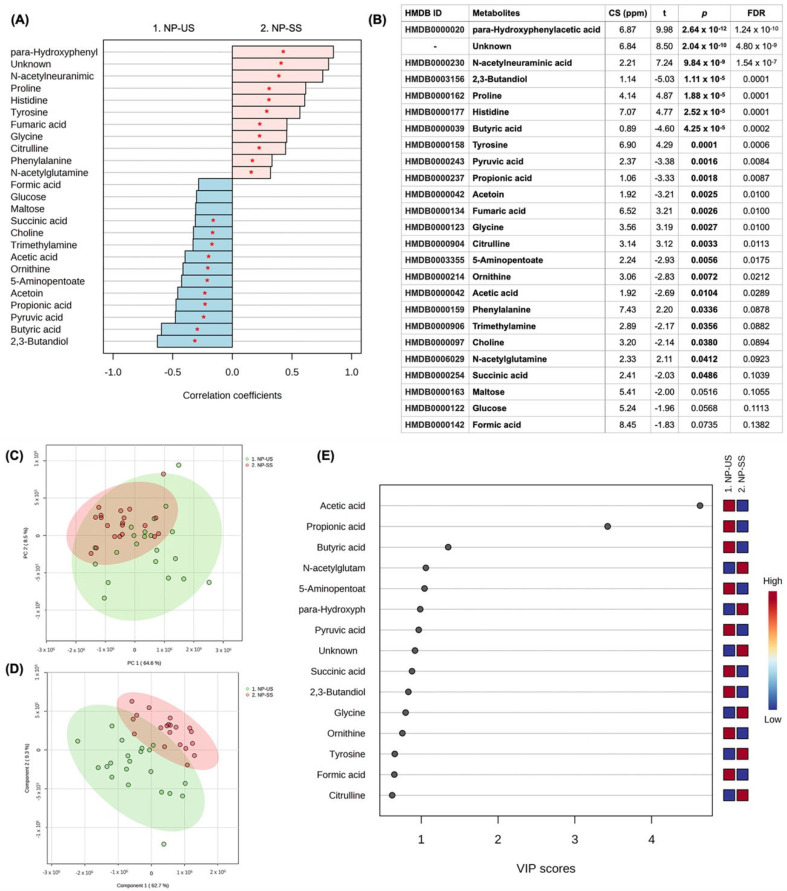
Univariate and multivariate analysis among US and SS samples from the NP group. (**A**) Correlation pattern. Red stars indicate *p* < 0.05. (**B**) Univariate analysis (*t* test); CS (ppm): chemical shift. (**C**) Score plot of PCA. (**D**) Score plot of the 1st and 2nd components of PLS-DA. (**E**) VIP scores of PLS-DA.

For the comparison between US and SS from the NWP group, 47 metabolites were identified, and 17 were significant in the univariate analysis (7 higher in the US and ten higher in SS). Butyric acid (*p* = 8.07 × 10^−6^), 5-Aminopentoate (*p* = 0.0009), Ornithine (*p* = 0.0028), Propionic acid (*p* = 0.0029), Acetic acid (*p* = 0.0069), Trimethylamine (*p* = 0.0273), and Succinic acid (*p* = 0.0482) were in higher concentration in US samples when compared to SS, while para-Hydroxyphenylacetic acid (*p* = 5.91 × 10^−14^), *N*-acetylneuraminic acid (*p* = 1.34 × 10^−14^), Histidine (*p* = 1.22 × 10^−8^), Proline (*p* = 3.61 × 10^−8^), Fumaric acid (*p* = 3.75 × 10^−5^), Tyrosine (*p* = 0.0027), Citrulline (*p* = 0.0053), *N*-acetyl glutamine (*p* = 0.0149), and Glycine were in higher concentrations in SS samples. One unidentified metabolite was higher in SS (Figure 4A,B). Multivariate analysis can be found in Figure 4C–E, highlighting the score plot of PCA (C), the score plot between the first two principal components of PLS-DA (D) (ACC = 0.96; R^2^ = 0.91; Q^2^: 0.77), and VIP scores based on PLS-DA (E). VIP scores of the PLS-DA indicated that Acetic acid, Propionic acid, 5-Aminopentoate, Butyric acid, Ethanol, Succinic acid, Pyruvic acid, Glucose, Ornithine, and Maltose were in higher concentration in US samples compared to SS, while *N*-acetyl glutamine, para-Hydroxyphenylacetic acid, and *N*-acetylneuraminic acid were in higher concentration in SS samples. Appendix A shows the concentration differences between US and SS in the NWP group for those metabolites that were highlighted in the VIP score and also presented *p* < 0.05 in the univariate analysis.

**Figure 4 jpm-13-01123-f004:**
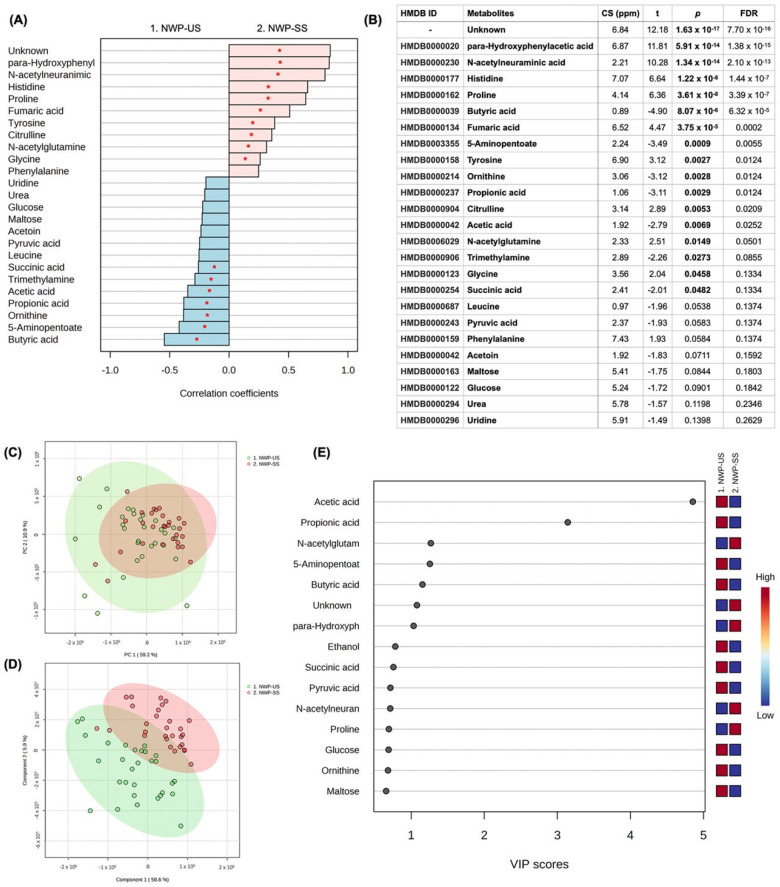
Univariate and multivariate analysis among US and SS samples from the NWP group. (**A**) Correlation pattern. Red stars indicate *p* < 0.05. (**B**) Univariate analysis (*t* test); CS (ppm): chemical shift. (**C**) Score plot of PCA. (**D**) Score plot of the 1st and 2nd components of PLS-DA. (**E**) VIP scores of PLS-DA.

Table 2 shows the metabolites in elevated concentrations in US or SS samples for each group, according to the univariate and/or multivariate analysis. Metabolites commonly found in all groups in elevated concentration in US samples were 5-Aminopentoate, Acetic acid, Butyric acid, Propionic acid, Pyruvic acid, and Succinic acid. Metabolites commonly found in all groups that were in elevated concentration in SS were Citrulline, Fumaric acid, Histidine, *N*-acetyl glutamine, *N*-acetylneuraminic acid, para-hydroxyphenylacetic acid, Proline, Tyrosine. Figure 5A,B refers to the Metabolic Enrichment Set Analysis. It shows the main metabolic pathways (I) and subclasses of metabolites (II) of those metabolites commonly found in all groups in US (A) and SS (B) samples, according to the univariate and multivariate analysis.

Regarding the metabolites commonly found in all groups that were in elevated concentration in the US, the main pathways indicated by MSEA were Butyrate Metabolism, Citric Acid Cycle, Amino Sugar Metabolism, Fatty Acids Biosynthesis, Pyruvate Metabolism, Glutamate Metabolism, and Warburg Effect (Figure 5A(I)). The main subclasses of the metabolites were Saturated Fatty Acids, Straight chain Fatty Acids, and Short-chain acids and derivates (Figure 5A(II)). Regarding the metabolites commonly found in all groups that were in elevated concentration in SS, the main pathways indicated by MSEA were Arginine and Proline Metabolism, Tyrosine Metabolism, Phenylalanine and Tyrosine Metabolism, Urea Cycle, Aspartate Metabolism, and Methylhistidine Metabolism (Figure 5B(I)). The main subclasses of the metabolites were Amino acids and *N*-acetylneuraminic acids (Figure 5B(II)).

**Figure 5 jpm-13-01123-f005:**
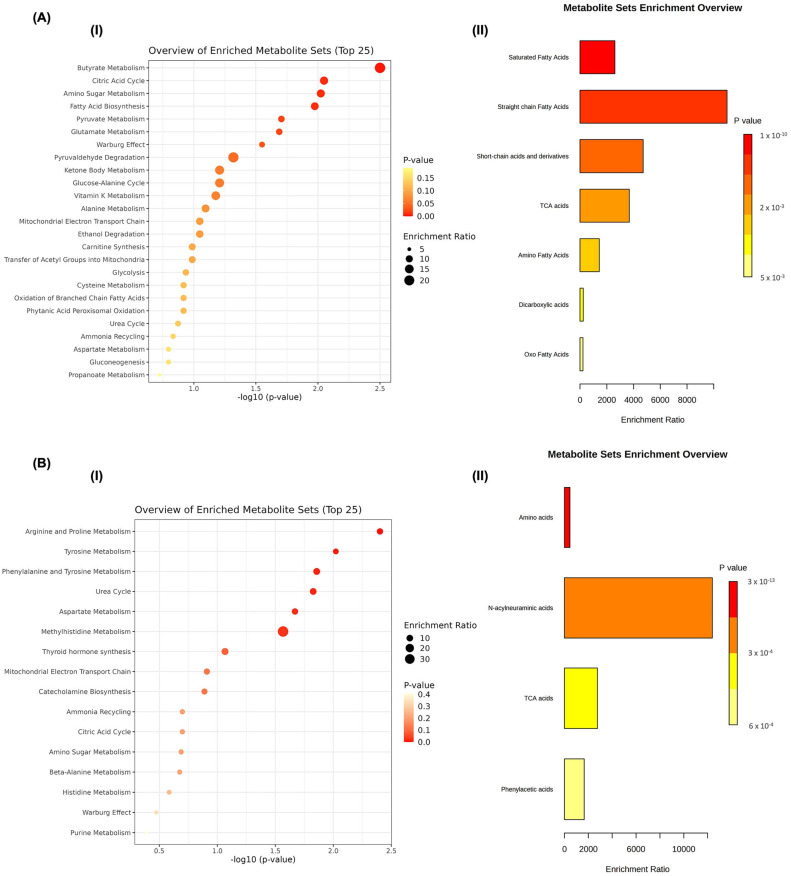
Metabolic Enrichment Set Analysis showing the main metabolic pathways (**I**) and subclasses of metabolites (**II**) of those metabolites in common among all groups in US (**A**) and SS (**B**) samples, according to the univariate and multivariate analysis.

## 4. Discussion

The main results of this study revealed that among the metabolites commonly present in all groups, 5-Aminopentoate, Acetic acid, Butyric acid, Propionic acid, Pyruvic acid, and Succinic acid were the metabolites that were in elevated concentrations in unstimulated saliva compared to stimulated saliva samples. These metabolites were mainly related to the following pathways: Butyrate Metabolism, Citric Acid Cycle, Amino Sugar Metabolism, Fatty Acids Biosynthesis, Pyruvate Metabolism, Glutamate Metabolism, and Warburg Effect. 

It is essential to point out that the main aim of this study was to compare the differences in the metabolic profile between unstimulated and stimulated saliva samples from pregnant women with obesity and/or periodontitis, and not to compare the results between these groups, which can be found in our previous report [27]. But as a supplement, we performed a metabolomic analysis comparing groups for stimulated saliva samples (see Appendix A, not published before). In summary, according to the univariate analysis, OP and NP groups showed higher concentrations of butyric acid, isovaleric acid, leucine, valine, isoleucine, propionic acid, acetic acid, trimethylamine, ornithine, and phenylalanine. Besides the higher levels of propionic acid, butyric acid, and acetic acid, the OP and NP groups showed lower concentrations of proline, *N*-acetylglutamine acid, citric acid, pyruvic acid, lactate, *N*-acetylneuraminic acid, and galactose, according to the multivariate analysis. The three top pathways related to the metabolites in higher concentration in OP and NP groups were: Valine, leucine, and isoleucine degradation; Fatty acid biosynthesis; and Propanoate metabolism. The five top pathways related to the metabolites in lower concentration in OP and NP groups were: the Warburg effect; the Transfer of acetyl groups into mitochondria; the Citric acid cycle, Amino sugar metabolism, and Gluconeogenesis. Although some differences were found in the intergroup comparison for SS samples compared to the intergroup comparison for US samples [27], stimulated saliva collection seems adequate in pregnant women with/without obesity and periodontitis, demonstrating similar metabolic pathways to unstimulated saliva samples. The main differences are discussed in detail below.

The submandibular gland contributes 65–70% of the total volume of unstimulated saliva, while the parotid and sublingual glands contribute 20% and 8%, respectively [38,39]. Stimulated saliva, in turn, is mainly produced by the parotid glands upon mechanical (i.e., using gums or chemical products—citric acid), pharmacological or sensory (i.e., taste and smell) stimulus [38]. So, after stimulation with citric acid, the contribution to the whole saliva from the parotid salivary gland doubles from 25% to 50% [40]. A previous study highlighted that acetate, acetone, alanine, formate, glycine, lactate, propionate, taurine, glucose, succinate, and dimethylamine were diluted in chemically stimulated saliva samples from healthy non-smoker male subjects when compared to the unstimulated saliva samples [39]. Our results are partially in line with these findings by Takeda et al. (2009) [39] since we revealed that acetic acid, propionic acid, and succinic acid were commonly found in all groups being in elevated concentration in the unstimulated saliva samples (Table 2). Formate, in turn, was found to be in higher concentration only in the US samples from periodontitis cases (OP and NP groups) (Figure 1 and Figure 3, Table 2), with no difference between US and SS samples from OWP and NWP groups. In this study, glucose also presented higher concentrations in the US than SS samples for OP, OWP, and NWP groups but not for the NP group. It is essential to highlight that acetic and propionic acids were previously found in the unstimulated saliva samples from individuals with the most severe periodontal parameters [27]. It is hypothesized that bacterial species, such as *H. parainfluenzae, N. sicca, Eubacterium, Propionibacterium, Arachnia, and Veillonella*, metabolize lactate to acetate and propionate, justifying their higher levels in unstimulated saliva samples. Thus, unstimulated saliva samples seem to be a better approach to identifying these metabolites related to periodontitis in pregnant women.

Our findings are also in accordance with the results by Neyraud et al. (2013) regarding the elevated concentration of propionate in unstimulated saliva samples. Nevertheless, in contrast with our findings, Neyraud et al. highlighted an over-representation of acetate in stimulated saliva [41]. They justified that as acetate is one the main organic acids found in biofilm, the stimulation through chewing a parafilm may have altered and released organic acids from the biofilm. However, we used the same saliva stimulation method in this study, and acetic acid was commonly found in all groups in elevated concentrations in the unstimulated saliva samples. More recently, Nam et al. (2023) also showed that from the 51 metabolites identified, 15 metabolites (organic acids, purine metabolites, choline and its metabolites, taurine, and *N*-acetylcadaverine) showed significant differences in levels between unstimulated saliva and stimulated saliva, being overexpressed when saliva was collected by mastication [42].

As aforementioned, with saliva stimulation, there is an expressive contribution of parotid glands in the content of saliva. Therefore, metabolites already present specifically in the parotid saliva are expected to be over-represented in stimulated saliva. In this study, we found an overexpression of proline and *N*-acetyl groups (*N*-acetyl glutamine, and *N*-acetylneuraminic) in stimulated saliva samples from all groups. Neyraud et al. found overexpression of proline, glutamine, lysine, and *N*-acetyl groups. These amino acids may directly result from the presence of proteins secreted by parotid glands, especially proline-rich proteins (PRPs), which account for 70% of the proteins secreted by these glands [41]. Yet, as supported by Neyraud et al., the overexpression of these amino acids and *N*-acetyl groups may be explained because the PRPs are intrinsically unstructured and consequently more sensitive to proteolysis occurring in the glands or the oral cavity [41]. Therefore, the over-representation of glutamine, lysine, proline, and *N*-acetyl groups in stimulated saliva may reflect the higher proportion of PRPs degradation products [41]. In our previous study [27], salivary levels of *N*-acetylneuraminic acid were positively correlated with periodontal parameters. This metabolite has been investigated as a potential biomarker of periodontitis since it may have a regulatory role in immunological processes, preventing oxidative stress and removing reactive oxygen species [43]. It has been hypothesized that the *N*-acetylneuraminic is synthesized as an acute phase protein to limit injury and encourage healing in individuals with periodontitis. In that context, analyzing the stimulated saliva sample could be positive to better understanding the role of the *N*-acetylneuraminic as a potential inflammatory biomarker in periodontitis cases.

In contrast with our results, Okuma et al. (2017) found the concentrations of 44 metabolites in stimulated saliva were significantly higher than those identified in unstimulated saliva, clustering the amino acids, for instance, into three groups: (1) histidine, phenylalanine, tryptophan, and arginine; (2) glycine, proline, asparagine; and (3) glutamine, serine, valine, leucine, isoleucine, threonine, lysine, alanine, and glutamic acid [44]. Interestingly, in this study, not only proline and *N*-acetyl groups were overexpressed in SS as aforementioned, but also histidine, tyrosine, fumaric acid, and citrulline were metabolites commonly identified in elevated concentration in all groups (Table 2).

The pathway analysis was shown by Okuma et al. [44]. identified the upregulation of the urea cycle and synthesis and degradation pathways of glycine, serine, cysteine, and threonine related to those 44 metabolites upregulated in stimulated saliva samples from healthy subjects. In this study, among the metabolites commonly found in all groups which were overregulated in US samples, the main pathways were the Butyrate Metabolism, Citric Acid Cycle, Amino Sugar Metabolism, Fatty Acids Biosynthesis, Pyruvate Metabolism, Glutamate Metabolism, and Warburg Effect (Figure 5A). On the other hand, the pathway analysis of metabolites overexpressed in SS samples of all groups from this study was similar to the pathways highlighted by Okuma et al. since we identified that those metabolites were mainly related to Arginine and Proline Metabolism, Tyrosine Metabolism, Phenylalanine and Tyrosine Metabolism, Urea Cycle, Aspartate Metabolism, and Methylhistidine Metabolism (Figure 5B).

Pyruvic acid, butyric acid and succinic acid were elevated in unstimulated saliva samples from all groups in this study (Figure 1, Figure 2, Figure 3, Figure 4 and Table 2). When we compared the four groups previously [27], these metabolites were related to the worst periodontal parameters in the sample. Therefore, these metabolites were previously linked to periodontitis, specifically disease-related microbiota. Two mechanisms justify the lower levels of pyruvic acid in periodontitis cases, as it is demonstrated in the pathway analysis of this study (Figure 5). Firstly, pyruvic acid represents the starting point of the citric acid cycle, which is upregulated in periodontal ligament cells during infection. Secondly, this metabolite is the principal substrate of L-lactate dehydrogenase, which exhibits increased activity during periodontal inflammation [27,45]. The increase in the levels of butyric acid and succinic acid is an essential indicator of the growth of pathogenic subgingival microorganisms (such as *Porphyromonas*, *Prevotella*, and *Fusobacterium* species) and the progression of periodontal tissue destruction [20,21,27]. Yet, the complex interplay between the host immune system and dysbiotic microflora is also reflected by changes in the salivary levels of butyrate and succinate [45]. *Porphyromonas gingivalis*-infected periodontal ligament cells showed high levels of succinate in a previous study [46], directly associated with energetic stress.

This study has some limitations. First, it is impossible to establish a cause-and-effect relationship between the studied outcomes, considering the cross-sectional design of the present study. Thus, a longitudinal study with a larger sample could deepen understanding of the salivary metabolic profile related to obesity and periodontitis. Ideally, future studies should assess the salivary metabolic profile before conception, throughout the different trimesters of pregnancy and after delivery. Still, studies evaluating the role of oral microbiota in the salivary metabolic profile of pregnant women with obesity and/or periodontitis are needed. Finally, individuals with different stages of periodontitis (I, II and III) were allocated to the same groups in this study (both for those with obesity and normal BMI). Stage I periodontitis refers to an initial form of the disease that may even resemble more advanced stages of gingivitis, a fact that may result in a difficult interpretation of the salivary metabolic profile related to the investigated outcomes. Therefore, future studies should be conducted isolating all stages of periodontal disease (gingivitis and periodontitis) in different groups to obtain a segmented understanding of the salivary metabolic profile considering the different levels of diseases. Despite the limitations, this study is relevant for demonstrating differences in the metabolic profile between unstimulated and stimulated saliva samples in the target population. 

In conclusion, there are some differences in the metabolic profile of unstimulated and stimulated saliva samples from pregnant women with/without obesity and periodontitis. Metabolites commonly found in all groups that were in elevated concentration in US samples showed Butyrate Metabolism, Citric Acid Cycle, Amino Sugar Metabolism, Fatty Acids Biosynthesis, Pyruvate Metabolism, Glutamate Metabolism, and Warburg Effect as the main metabolic pathways, while metabolites commonly found in all groups that were in elevated concentration in SS had Arginine and Proline Metabolism, Tyrosine Metabolism, Phenylalanine and Tyrosine Metabolism, Urea Cycle, Aspartate Metabolism, and Methylhistidine Metabolism as the main metabolic pathways. Although some differences were found in the intergroup comparison for SS samples compared to the intergroup comparison for US samples, stimulated saliva collection seems adequate in pregnant women with/without obesity and periodontitis, demonstrating similar metabolic pathways to unstimulated saliva samples.

## Figures and Tables

**Table 1 jpm-13-01123-t001:** Comparison among groups for anthropometric and periodontal variables.

	OP (n = 20) Median [1st–3rd Quartiles] Mean ± SD	OWP (n = 27) Median [1st–3rd Quartiles] Mean ± SD	NP (n = 20) Median [1st–3rd Quartiles] Mean ± SD	NWP (n = 29) Median [1st–3rd Quartiles] Mean ± SD	*p*
Pre-pregnancy BMI (kg/m^2^)	31.37 [30.02–35.08] A	32.74 [30.09–36.38] A	22.76 [20.11–24.79] B	23.23 [22.17–24.66] B	**<0.001 ***
Gestational weight gain (kg)	7.25 ± 6.73	8.00 ± 5.60	10.67 ± 6.88	10.77 ± 5.26	0.105 ^†^
PPD (mm)	2.48 [2.25–2.65] A	2.03 [1.98–2.10] B	2.54 [2.42–2.78] A	2.07 [2.01–2.16] B	**<0.001 ***
CAL (mm)	2.48 [2.30–2.66] A	2.06 [1.99–2.11] B	2.54 [2.43–2.81] A	2.08 [2.02–2.16] B	**<0.001 ***
Periodontitis—n (%)		-		-	-
Stage I	11 (55%)		9 (45%)		
Stage II	8 (40%)		9 (45%)		
Stage III	1 (5%)		2 (10%)		
Stage IV	0		0		

BMI, body mass index; PPD, probing pocket depth; CAL, clinical attachment level; OP, pregnant women with obesity and periodontitis; OWP, pregnant women with obesity but without periodontitis; NP, pregnant women with normal BMI but with periodontitis; NWP, pregnant women with normal BMI and without periodontitis; SD, standard deviation; *p*, significance level; * Kruskal-Wallis (post hoc: Dunn); † One-way ANOVA (post hoc: Tukey); Different letters indicate differences between groups; bold values indicate *p* < 0.05.

**Table 2 jpm-13-01123-t002:** Metabolites in elevated concentrations in US or SS samples for each group, according to the univariate and/or multivariate analysis.

**Metabolites in Elevated Concentration in US Samples ***			
OP	OWP	NP	NWP
5-Aminopentoate	5-Aminopentoate	5-Aminopentoate	5-Aminopentoate
Acetic acid	Acetic acid	Acetic acid	Acetic acid
Butyric acid	Butyric acid	Butyric acid	Butyric acid
Propionic acid	Propionic acid	Propionic acid	Propionic acid
Pyruvic acid	Pyruvic acid	Pyruvic acid	Pyruvic acid
Succinic acid	Succinic acid	Succinic acid	Succinic acid
2,3-Butanediol	Acetoin	2,3-Butanediol	Ethanol
Formic acid	2,3-Butanediol	Acetoin	Glucose
Glucose	Glucose	Choline	Maltose
Maltose	Lactate	Formic acid	Ornithine
	Maltose	Ornithine	Trimethylamine
	Ornithine		
	Trimethylamine		
**Metabolites in elevated concentration in SS samples ***			
OP	OWP	NP	NWP
Citrulline	Citrulline	Citrulline	Citrulline
Fumaric acid	Fumaric acid	Fumaric acid	Fumaric acid
Histidine	Histidine	Histidine	Histidine
***N***-acetylglutamine	*N*-acetylglutamine	*N*-acetylglutamine	*N*-acetylglutamine
***N***-acetylneuraminic acid	*N*-acetylneuraminic acid	*N*-acetylneuraminic acid	*N*-acetylneuraminic acid
***p***-hydroxyphenylacetic acid	*p*-hydroxyphenylacetic acid	*p*-hydroxyphenylacetic acid	*p*-hydroxyphenylacetic acid
Proline	Proline	Proline	Proline
Tyrosine	Tyrosine	Tyrosine	Tyrosine
		Glycine	Glycine
		Phenylalanine	

* Metabolites in higher concentration that were highlighted in multivariate analysis (VIP scores) and/or presented *p* < 0.05 in the univariate analysis; OP, pregnant women with obesity and periodontitis; OWP, pregnant women with obesity but without periodontitis; NP, pregnant women with normal BMI but with periodontitis; NWP, pregnant women with normal BMI and without periodontitis.

## Data Availability

The data presented in this study are available within this article and in Appendix A. Also, data will be made accessible to other researchers upon request.

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
