# Peer review of "Comparison of the Metabolic Profile between Unstimulated and Stimulated Saliva Samples from Pregnant Women with/without Obesity and Periodontitis"

_jpm, 2023, doi:10.3390/jpm13071123_

Round 1

Reviewer 1 Report

Title: 

Overall, the title and abstract provide a clear overview of the study, its objectives, methods, and preliminary findings.

Abstract: 

The abstract lacks specific information on the statistical analyses performed to compare the metabolic profiles between the different groups.

The abstract could provide more context regarding the potential clinical implications of the observed differences in the metabolic profiles.

Introduction:

The introduction of the manuscript has several shortcomings that could be improved upon.

Lack of Clear Objectives: The introduction does not explicitly state the specific objectives or research questions that the study aims to address. It is important to clearly outline the purpose of the study at the beginning of the introduction to provide a roadmap for the readers and establish the focus of the research.

Insufficient Background Information: While the introduction briefly touches upon the concept of periodontal medicine and its interrelationship with systemic changes in the body, it lacks a comprehensive literature review that establishes the current state of knowledge in the field. Providing a more thorough background on the existing research, theories, and controversies surrounding the topic would help contextualize the study and highlight the research gap it aims to fill.

Limited Justification: The introduction briefly mentions the increased number of diseases and conditions potentially related to periodontal diseases and highlights the associations between periodontitis, pregnancy, and obesity. However, it does not provide a strong justification for the importance or relevance of studying the metabolic profiles of stimulated and unstimulated saliva in pregnant women with/without obesity and periodontitis. It would be beneficial to expand on the clinical significance and potential implications of investigating these metabolic profiles.

Incomplete Citation of Previous Studies: The introduction cites previous studies that have investigated proteomic and metabolomic profiles of unstimulated saliva in pregnant women with obesity and periodontitis, but it does not provide proper citations (only references [8,9]). It is important to provide complete and accurate citations to allow readers to access and refer to the relevant literature.

Lack of Flow and Cohesion: The introduction appears to be a collection of fragmented statements without a clear logical flow. It would benefit from restructuring the information and organizing it into cohesive paragraphs that gradually build up to the study's objectives and rationale.

Absence of a Hypothesis: The introduction briefly mentions a null hypothesis without elaborating on the specific hypotheses or expected outcomes of the study. Including a clear hypothesis or research question would help guide the study and provide a framework for interpreting the results.

Methodology:

There are several shortcomings and areas that could be improved.

Lack of detailed information on the sampling method: The methodology section states that the sampling method, as well as the inclusion and exclusion criteria, were based on a previous study by Foratori-99 Junior et al. (2022). However, it would be helpful to provide more specific details about the sampling method used in the current study. Information about how participants were selected and recruited, any potential sources of bias in participant selection, and the representativeness of the sample would enhance the transparency and reliability of the study.

Limited justification for sample size: The methodology mentions that the sample size was based on previous reports. However, it would be beneficial to provide a more detailed explanation or a power analysis justifying the chosen sample size. This would help readers assess whether the sample size is adequate for detecting meaningful differences and generalizing the findings to the target population.

Incomplete description of data collection procedures: The methodology provides a brief overview of the data collection procedures, including periodontal parameter measurements and saliva collection. However, important details are missing. For example, information about the calibration of the dentist who collected the periodontal parameters (such as calibration exercises or intra-examiner reliability) would enhance the validity and reliability of the measurements. Similarly, details about the instructions given to participants regarding saliva collection, the use of standardized protocols, and any potential sources of variability or error in the collection process would be valuable.

Insufficient information on NMR analysis: The methodology briefly mentions the use of 1H-NMR analysis for metabolite profiling of saliva samples. However, critical details are missing, such as the specific parameters and settings used during the spectral acquisition, the duration of the NMR experiments, and any quality control measures taken during the analysis. Providing this information would allow readers to assess the robustness and reproducibility of the metabolomic data.

Limited information on statistical analysis: The methodology mentions the use of various statistical tests, including ANOVA, Kruskal-Wallis, t-test, and correlation analysis. However, it lacks details regarding the specific variables analyzed, the significance thresholds used, and the correction for multiple testing. Additionally, it would be helpful to mention the software or statistical package used for the analyses.

Insufficient information on bioinformatics analysis: The methodology briefly mentions the use of PCA, PLS-DA, VIP scores, and metabolite set enrichment analyses. However, there is a lack of details on the specific settings, parameters, and software used for these analyses. Providing more information about the bioinformatics analysis would enable readers to understand and reproduce the results more accurately.

Missing information on potential confounders and adjustments: The methodology does not mention whether any potential confounding variables were considered or adjusted for in the analyses. Given the complexity of the associations being investigated (e.g., obesity, periodontitis, and metabolic profiles), it is important to address potential confounders and control for them in the statistical analysis to obtain more accurate and reliable results.

Lack of data availability statement: In the current era of open science, it is important to include a statement about data availability, indicating whether the data will be made accessible to other researchers upon request. This promotes transparency, reproducibility, and further scientific collaboration.

Discussion:

The study successfully compared the metabolic profiles of unstimulated and stimulated saliva samples from pregnant women with obesity and/or periodontitis, providing valuable insights into the differences between these two sample types.

The identification of metabolites commonly present in all groups, such as 5-Aminopentoate, Acetic acid, Butyric acid, Propionic acid, Pyruvic acid, and Succinic acid, and their association with specific metabolic pathways (Butyrate Metabolism, Citric Acid Cycle, Amino Sugar Metabolism, Fatty Acids Biosynthesis, Pyruvate Metabolism, Glutamate Metabolism, and Warburg Effect) provides a comprehensive understanding of the salivary metabolic profile in relation to the studied conditions.

The study supports previous research findings that highlight the potential of unstimulated saliva samples in identifying metabolites related to periodontitis in pregnant women, particularly the elevated concentrations of acetic acid, propionic acid, and succinic acid in unstimulated saliva.

The findings regarding the overexpression of proline and N-acetyl groups in stimulated saliva samples suggest their origin from proteins secreted by the parotid glands, specifically proline-rich proteins (PRPs), which have been associated with periodontal health.

The study highlights the potential role of N-acetylneuraminic acid as a potential inflammatory biomarker in periodontitis, and the importance of analyzing stimulated saliva to better understand its function in periodontal diseases.

The pathway analysis results align with previous studies, supporting the involvement of specific metabolic pathways in periodontitis and obesity, and reinforcing the value of unstimulated saliva samples in identifying metabolites associated with these conditions.

Author Response

July 07th, 2023

We would like to thank the editors and reviewers for the caution and effort at revising our manuscript entitled “Comparison of the metabolic profile between unstimulated and stimulated saliva samples from pregnant women with/without obesity and periodontitis”. The reviewers’ positive opinions regarding the merit of the present study were enriching so that we could adapt the manuscript in order to make it suitable for publication in this respectful journal. In addition, the authors point out that the quality of the review performed by the reviewers reflects the knowledge and mastery of these professionals in this research field, which contributed to improve our manuscript.

   We hope to have answered all the questions raised by the editor and reviewers, but we are available for any other questions.

­Title: 

Overall, the title and abstract provide a clear overview of the study, its objectives, methods, and preliminary findings.

Answer: Thank you for your comment. We kept the manuscript’s title as the original.

Abstract: 

The abstract lacks specific information on the statistical analyses performed to compare the metabolic profiles between the different groups.

Answer: As the main point of the study was to compare the metabolic profile of the stimulated and unstimulated saliva samples, we decided to include the intergroups analysis for stimulated saliva samples as a Supplementary File (see Supplementary File S1). Also, to avoid a very long abstract (there is a limit of characters according to the journal’s rules), we did not include in the abstract information regarding the intergroup comparations for stimulated saliva samples. Nonetheless, in the supplementary file S1 we detailed the univariate and multivariate analyses comparing each group, and we also pointed out the main metabolic pathways of the metabolites that were in higher or lower concentration in the main groups (OP and NP groups). Also, we included a paragraph in the result section summarizing these results.

We hope to have met the reviewer’s expectations. Thank you.

Abstract: 

The abstract could provide more context regarding the potential clinical implications of the observed differences in the metabolic profiles.

Answer: Thank you for your suggestion. We included a sentence at the end of the abstract that highlight the clinical implications of our results, as follows: “Although some differences were found between unstimulated and stimulated saliva samples from pregnant women with/without obesity and periodontitis, stimulated saliva collection seems to be adequate, demonstrating similar metabolic pathways to unstimulated saliva samples when groups are compared.

Introduction:

The introduction of the manuscript has several shortcomings that could be improved upon.

Lack of Clear Objectives: The introduction does not explicitly state the specific objectives or research questions that the study aims to address. It is important to clearly outline the purpose of the study at the beginning of the introduction to provide a roadmap for the readers and establish the focus of the research.

Answer: Thank you for your comment. We restructured the first paragraph of the introduction section to meet your suggestion. Therefore, we presented the aim of our study at the beginning of introduction section.

Modified text:Several studies have investigated the epidemiological association of obesity and periodontitis in pregnant women [1], nonetheless, the pathophysiological understanding of the association of these outcomes is not clear. Salivary analysis has been widely used to obtain an understanding of the individual's biological mechanisms related to health or diseases. In addition to being an easily accessible and non-invasive method, saliva has a variety of proteins and metabolites that can be studied as potential biomarkers of systemic and oral diseases [2]. In some situations, it may be difficult to collect unstimulated saliva from individuals due to systemic or oral changes that impair salivary flow, as occurs in obesity, pregnancy, and periodontitis. Then, it is important to methodologically assess whether there are differences in the components in samples of unstimulated and stimulated saliva. Therefore, this study aimed to compare the metabolic profile of unstimulated and stimulated saliva samples from pregnant women with/without obesity and periodontitis by Proton Nuclear Magnetic Resonance (1H-NMR).

Introduction:

Insufficient Background Information: While the introduction briefly touches upon the concept of periodontal medicine and its interrelationship with systemic changes in the body, it lacks a comprehensive literature review that establishes the current state of knowledge in the field. Providing a more thorough background on the existing research, theories, and controversies surrounding the topic would help contextualize the study and highlight the research gap it aims to fill.

Limited Justification: The introduction briefly mentions the increased number of diseases and conditions potentially related to periodontal diseases and highlights the associations between periodontitis, pregnancy, and obesity. However, it does not provide a strong justification for the importance or relevance of studying the metabolic profiles of stimulated and unstimulated saliva in pregnant women with/without obesity and periodontitis. It would be beneficial to expand on the clinical significance and potential implications of investigating these metabolic profiles.

Lack of Flow and Cohesion: The introduction appears to be a collection of fragmented statements without a clear logical flow. It would benefit from restructuring the information and organizing it into cohesive paragraphs that gradually build up to the study's objectives and rationale.

Answer: Thank you for your comment. We agree with the reviewer that our introduction section needed to be complemented and better justified. Therefore, we performed several changes throughout the introduction. We included some paragraphs to have a better background information, also we changed the sequence of our paragraphs to reach a better rationale, flow, and cohesion. Finally, we provided some sentences that better justify our research question and the purpose of our study. We hope to have met the reviewer expectation. However, we make ourselves available for future modifications.

Introduction:

Incomplete Citation of Previous Studies: The introduction cites previous studies that have investigated proteomic and metabolomic profiles of unstimulated saliva in pregnant women with obesity and periodontitis, but it does not provide proper citations (only references [8,9]). It is important to provide complete and accurate citations to allow readers to access and refer to the relevant literature.

Answer: Thank you for this suggestion. We agree with the reviewer. We included several new references in the introduction section to support our background.

Introduction:

Absence of a Hypothesis: The introduction briefly mentions a null hypothesis without elaborating on the specific hypotheses or expected outcomes of the study. Including a clear hypothesis or research question would help guide the study and provide a framework for interpreting the results.

Answer: Thank you for this suggestion. We restructured our hypotheses based on the research question. We included the research question as well. Similarly, we elaborated an alternative hypothesis. Thank you so much.

Modified text:Therefore, our research question was “are there differences in the metabolic profile when unstimulated and stimulated saliva samples from pregnant women with/without obesity and periodontitis are compared?”. The null hypothesis adopted in this study was that there are no differences in the metabolic profile of stimulated saliva compared to non-stimulated saliva samples from pregnant women with/without obesity and periodontitis. The alternative hypothesis, in turn, was that salivary stimulation significantly altered salivary metabolites related to obesity and/or periodontitis when compared to unstimulated saliva, therefore not being the primary method of choice for metabolomic analysis.

Methodology:

There are several shortcomings and areas that could be improved.

Lack of detailed information on the sampling method: The methodology section states that the sampling method, as well as the inclusion and exclusion criteria, were based on a previous study by Foratori-99 Junior et al. (2022). However, it would be helpful to provide more specific details about the sampling method used in the current study. Information about how participants were selected and recruited, any potential sources of bias in participant selection, and the representativeness of the sample would enhance the transparency and reliability of the study.

Answer: Thank you for this suggestion. We had only referenced this subsection of sampling method to avoid a high percentage of plagiarism with previous studies from our team. But we restructured that information accordingly to give a more detailed description about the sampling method. Thank you so much.

Modified text:Inclusion criteria adopted in this study were: pregnant women, during the 3rd trimester of pregnancy, aged 18–40 years old, with regular registration in the public health system in Brazil and with regular follow-up with an obstetrician (at least one prenatal medical visit per trimester). Women should present adequate cognitive function during pregnancy, without impairments that required absolute rest. In contrast, the team adopted the following exclusion criteria before the beginning of recruitment: twin pregnancy, individuals who had neuromotor impairments, arterial hypertension during pregnancy (blood pressure   140/90 mmHg), gestational diabetes mellitus (hyperglycemia:   92 mg/dL—fasting level;  180 mg/dL—after 1 h; and  153 mg/dL—after 2 h); malnutrition (BMI < 18.50 kg/m2), overweight (BMI between 25.00 kg/m2 and 29.99 kg/m2), confirmed or suspected diagnosis of SARS-CoV-2 infection at the moment of the first appointment, hyposalivation (< 0.25 mL/min flow rate), who were taking antibiotics or had taken antibiotics during pregnancy, who were taking any medication that could interfere with periodontal status and/or salivary flow (e.g., immunosuppressive drugs, anticonvulsants or calcium channel blockers), who were under orthodontic treatment or any dental treatment with another professional, who had multiple tooth loss (more than two teeth per hemiarch), who had stage IV periodontitis, self-reported systemic disease besides obesity, and users of alcohol/tobacco/illicit drugs [27].

Methodology:

Limited justification for sample size: The methodology mentions that the sample size was based on previous reports. However, it would be beneficial to provide a more detailed explanation or a power analysis justifying the chosen sample size. This would help readers assess whether the sample size is adequate for detecting meaningful differences and generalizing the findings to the target population.

Answer: Thank you for this comment.

The question regarding the calculation of the sample size is a recurring discussion raised in the reviews of articles that adopt the metabolomic analysis. It is important to observe that we adopted restrict inclusion criteria for the sample selection. Being women, 18-40 years-old, during the 3rd trimester of pregnancy, with or without obesity, with or without periodontitis, without other systemic impairments (to avoid bias) are examples of criteria adopted in this study to obtain the most homogeneous sample. Considering our rigorous inclusion criteria, it was expected we would not have a large sample, especially in the group of pregnant women with obesity and periodontitis. However, we emphasize that all pregnant women followed up by the health team from healthcare units in Bauru during the period of recruitment, and who met our criteria, were included in this study. In addition, we are in accordance with previous studies that used the same protocol for metabolomics in respect of periodontitis, as referenced in the text. So, in metabolomic studies it is not common to perform the sample size calculation, and the published literature presents other studies with this number (da Costa Rosa, de Almeida Neves et al. 2021, Fidalgo, Freitas-Fernandes et al. 2021, Freitas-Fernandes, Fidalgo et al. 2021, Loureiro, Ferreira et al. 2021).

Yet, it is important to point out that the main objective of our study was to determine the metabolomic profile and pathway of unstimulated and stimulated saliva samples in pregnant women with/without obesity and with/without periodontitis. Therefore, this is an untargeted metabolomic approach. Considering that this study adopts an untargeted metabolomic approach, there is no specific outcome that may be used to calculate the “power of the test” in relation to the sample size. Adopting a single metabolite as an outcome to calculate the power of the test would not be correct because it would not represent the main objective of characterizing/mapping the metabolomic profile of the studied population. Thus, the outcome of this study refers to the mapping and characterization of saliva metabolites, taking into account obesity and periodontitis as outcomes.

References:

  • da Costa Rosa, T., A. de Almeida Neves, M. A. Azcarate-Peril, K. Divaris, D. Wu, H. Cho, K. Moss, B. J. Paster, T. Chen, L. B. Freitas-Fernandes, T. K. S. Fidalgo, R. Tadeu Lopes, A. P. Valente, R. R. Arnold and A. de Aguiar Ribeiro (2021). "The bacterial microbiome and metabolome in caries progression and arrest." Journal of Oral Microbiology 13(1): 1886748.
  • Fidalgo, T., L. B. Freitas-Fernandes, F. C. L. Almeida, I. P. R. Souza and A. P. Valente (2021). "Effect of antihistamine-containing syrup on salivary metabolites: an in vitro and in vivo study." Braz Oral Res 35: e032.
  • Freitas-Fernandes, L. B., T. K. S. Fidalgo, P. A. de Almeida, I. P. R. Souza and A. P. Valente (2021). "Salivary metabolome of children and adolescents under peritoneal dialysis." Clin Oral Investig 25(4): 2345-2351.
  • Loureiro, L. L., T. J. Ferreira, C. S. C. da Costa, T. K. S. Fidalgo, A. P. Valente and A. Pierucci (2021). "Impact of Precompetitive Training on Metabolites in Modern Pentathletes." Int J Sports Physiol Perform 17(3): 489-494.

Methodology:

Incomplete description of data collection procedures: The methodology provides a brief overview of the data collection procedures, including periodontal parameter measurements and saliva collection. However, important details are missing. For example, information about the calibration of the dentist who collected the periodontal parameters (such as calibration exercises or intra-examiner reliability) would enhance the validity and reliability of the measurements. Similarly, details about the instructions given to participants regarding saliva collection, the use of standardized protocols, and any potential sources of variability or error in the collection process would be valuable.

Answer: Thank you for this suggestion. We included a paragraph in the methodology section describing the calibration of the examiner. Thank you.

Modified text:To group individuals, pre-pregnancy nutritional status based on cut-off point of BMI proposed by World Health Organization (WHO) was considered. Periodontal status was based on measurements of probing pocket depth (PPD) and clinical attachment level (CAL)/attachment loss (AL), and then patients were classified as having or not periodontitis. All data collection was performed by one dentist previously calibrated by a gold standard examiner in epidemiological surveys, in order to ensure uniformity in the collection of data regarding the periodontal conditions. The kappa coefficient (kappa inter-examiner reliability = 0.92; kappa intra-examiner reliability = 0.95) was calculated based on periodontal diagnosis of approximately 10% of the sample (n = 10). A 15-day interval was observed between the examinations performed by the principal examiner dentist and the gold standard examiner because of periodontal alteration after the first examination. The examiner's training took place through lectures, study and discussion of didactic material, and demonstrative practical activity with non-pregnant individuals. This training took place to obtain fluidity in the service, correct application of the questionnaires, correct access to the system to obtain previous data (pre-pregnancy weight, information from previous prenatal medical consultations etc.), correct periodontal ex-amination to obtain the PPD and CAL/AL parameters, correct collection of saliva samples. The examiner was properly instructed to conduct the exams and saliva collection, learning to guide the patients correctly, supervise the entire saliva collection, in order to prevent the patients from talking during the process or swallowing the saliva produced. The examiner was also trained to supervise the collection of stimulated saliva, guiding the participants to chew the parafilm gum properly, also avoiding the risk of them swal-lowing the parafilm or providing support if they felt nauseous.

Methodology:

Insufficient information on NMR analysis: The methodology briefly mentions the use of 1H-NMR analysis for metabolite profiling of saliva samples. However, critical details are missing, such as the specific parameters and settings used during the spectral acquisition, the duration of the NMR experiments, and any quality control measures taken during the analysis. Providing this information would allow readers to assess the robustness and reproducibility of the metabolomic data.

Answer: All system parameters for metabolomic analysis were properly described in the manuscript, as follows: “Bruker Avance NEO 600 MHz equipped with a TCI Cryoprobe Prodigy (Bruker Biospin, Karlsruhe, Germany), operating at a proton frequency of 600.2 MHz at 298 K was used for spectral acquisition. 1H spectra were acquired using Carr-Purcell-Meiboom-Gill (CPMG) pulse sequence with water suppression by presaturation [34], with 32 scans, 65,536 points for the free induction decay (FID) during the acquisition, a spectra width of 20.8 ppm, an acquisition time of 2.62 s, and an inter-scan delay of 4 s. TSP peak (0 ppm) was used as internal reference/standard.”. These parameters are the same that were described in previous studies, including from our team. Thank you.

Methodology:

Limited information on statistical analysis: The methodology mentions the use of various statistical tests, including ANOVA, Kruskal-Wallis, t-test, and correlation analysis. However, it lacks details regarding the specific variables analyzed, the significance thresholds used, and the correction for multiple testing. Additionally, it would be helpful to mention the software or statistical package used for the analyses.

Answer: Thank you. Most part of this information is already detailed in the manuscript. We included the significance thresholds. Thank you for this consideration.

Also, at the bottom of table 1, we included symbols which identify which statistical test was used for each variable. That is the reason we did not mention each variable in the “statistical analysis” subsection, in order to avoid a very long manuscript. The statistical software is properly described: “IBM SPSS Version 25 (IBM Corp. Released 2017. IBM SPSS Statistics for Windows, Version 25.0. Armonk, NY: IBM Corp.).

Methodology:

Insufficient information on bioinformatics analysis: The methodology briefly mentions the use of PCA, PLS-DA, VIP scores, and metabolite set enrichment analyses. However, there is a lack of details on the specific settings, parameters, and software used for these analyses. Providing more information about the bioinformatics analysis would enable readers to understand and reproduce the results more accurately.

Answer: Thank you for this comment. We provided a more detailed information in the “statistical analysis and bioinformatics” subsection, as follows:

Modified text:For clinical parameters, data were presented as mean and standard deviation, me-dian and 1st-3rd quartiles, or percentages. Statistical analysis was performed with IBM SPSS Version 25 (IBM Corp. Released 2017. IBM SPSS Statistics for Windows, Version 25.0. Armonk, NY: IBM Corp.). Quantitative variables were firstly tested for normality using the Kolmogorov-Smirnov test. One-way ANOVA with Tukey's test as post-hoc were used for quantitative variables with normal distribution, while Kruskal-Wallis with Dunn's test as post-hoc were used for quantitative variables without normal distribution. A significance level of 5% was considered.

For metabolomic analysis, univariate analysis was handled through t test and cor-relations using the IBM SPSS software Version 25 (IBM Corp. Released 2017. IBM SPSS Statistics for Windows, Version 25.0. Armonk, NY: IBM Corp.) to evaluate whether the overall comparison was significant among US and SS for each group. Multivariate analysis was based on Principal Component Analysis (PCA) and Partial Least Square – Discriminant Analysis (PLS-DA) to obtain the predictive performance of the models; each model was evaluated for Q2, R2, and accuracy (ACC). For each model, 1000 permutations were performed. The variable importance in projection (VIP) scores were obtained based on PLS-DA for the determination of the relative abundances of the main 15 metabolites that contributed to the separation between US and SS. There are two important measures in PLS-DA: one is the variable importance in projection (VIP) and the other is the weighted sum of absolute regression coefficients. The colored boxes on the right of each figure of the results presented here (see figures below; Figure 1–4) indicate the relative concentrations of the corresponding metabolite in each subgroup under study (in this case, US and SS samples). All these multivariate analyses were handled on MetaboAnalyst 5.0 software (www.metaboanalyst.ca) (accessed on 20 September 2022) after sample normalization by sum, and Log transformation (base 10), and data scaling by Pareto scaling (mean-centered and divided by the square root of the standard deviation of each variable).

Metabolite set enrichment analyses (MSEA) were conducted for visualization and functional analysis of metabolites [27,37] on MetaboAnalyst 5.0 software (www.metaboanalyst.ca) (accessed on 20 September 2022). Only metabolites highlighted in the VIP score and that also presented p < 0.05 in the univariate analysis in common for each group were included in the MSEA, both for US and SS samples. In the MSEA for US samples, the metabolites included were 5-Aminopentoate, Acetic acid, Butyric acid, Propionic acid, Pyruvic acid, and Succinic acid. In the MSEA for SS samples, the metabolites included were Citrulline, Fumaric acid, Histidine, N-acetylglutamine, N-acetylneuraminic acid, para-hydroxyphenylacetic acid, Proline, and Tyrosine.

As a supplement, we performed metabolomic analysis with comparison between groups for stimulated saliva samples (see Supplementary File S1). For this, we adopted one-way ANOVA and Tukey test as post-hoc for the univariate analysis to compare the normalized concentrations of each metabolite (showing the mean concentration and its standard deviation). We also reported the multivariate analysis through the scores plot between the first two principal components (PCs) of sPLS-DA, and the loadings plot of component 1 from the sPLS-DA. As aforementioned, multivariate analysis was handled on MetaboAnalyst 5.0 software (www.metaboanalyst.ca) (accessed on 05 July 2023) after sample normalization by sum, Log transformation (base 10), and data scaling by Pareto scaling (mean-centered and divided by the square root of the standard deviation of each variable). MSEA were also conducted for visualization and functional analysis of metabolites in the intergroup comparations for stimulated saliva samples. Metabolites that were in higher or lower concentrations according to the univariate and multivariate analyses in the intergroup comparations of SS samples were properly included in the MSEA, using the MetaboAnalyst 5.0 software (www.metaboanalyst.ca) (accessed on 20 September 2022).

Methodology:

Missing information on potential confounders and adjustments: The methodology does not mention whether any potential confounding variables were considered or adjusted for in the analyses. Given the complexity of the associations being investigated (e.g., obesity, periodontitis, and metabolic profiles), it is important to address potential confounders and control for them in the statistical analysis to obtain more accurate and reliable results.

Answer: Thank you for this comment. At the beginning of methodology section, after the inclusion and exclusion criteria description, we include a sentence demonstrating the reason for adopting rigorous inclusion and exclusion criteria. We aimed to get the most homogenous sample possible in order to avoid any bias or confounders, or yet to minimize the variability of the sample in the metabolomic analysis. We followed a rigorous methodology from previous studies to ensure a more reliable metabolomic analysis based on the outcomes (obesity and periodontitis; and the differences among unstimulated and stimulated saliva samples).

Modified text:These rigorous inclusion and exclusion criteria were adopted to reach the most homogenous sample possible, avoiding any biases or confounders in the analysis of the met-abolic profile of unstimulated and stimulated saliva. Therefore, it was expected that by adopting these criteria and having a good sample pairing, we would avoid great variability in the metabolomic analysis, being able to have a better understanding of the role of the study outcomes (obesity and periodontitis).

Methodology:

Lack of data availability statement: In the current era of open science, it is important to include a statement about data availability, indicating whether the data will be made accessible to other researchers upon request. This promotes transparency, reproducibility, and further scientific collaboration.

Answer: Thank you for this suggestion. We included this sentence at the end of manuscript above acknowledgments. Thank you.

Modified text:Data Availability Statement: The data presented in this study are available within this article and in the Supplementary Files S1–S5. Also, data will be made accessible to other researchers upon request.

Discussion:

The study successfully compared the metabolic profiles of unstimulated and stimulated saliva samples from pregnant women with obesity and/or periodontitis, providing valuable insights into the differences between these two sample types.

The identification of metabolites commonly present in all groups, such as 5-Aminopentoate, Acetic acid, Butyric acid, Propionic acid, Pyruvic acid, and Succinic acid, and their association with specific metabolic pathways (Butyrate Metabolism, Citric Acid Cycle, Amino Sugar Metabolism, Fatty Acids Biosynthesis, Pyruvate Metabolism, Glutamate Metabolism, and Warburg Effect) provides a comprehensive understanding of the salivary metabolic profile in relation to the studied conditions.

The study supports previous research findings that highlight the potential of unstimulated saliva samples in identifying metabolites related to periodontitis in pregnant women, particularly the elevated concentrations of acetic acid, propionic acid, and succinic acid in unstimulated saliva.

The findings regarding the overexpression of proline and N-acetyl groups in stimulated saliva samples suggest their origin from proteins secreted by the parotid glands, specifically proline-rich proteins (PRPs), which have been associated with periodontal health.

The study highlights the potential role of N-acetylneuraminic acid as a potential inflammatory biomarker in periodontitis, and the importance of analyzing stimulated saliva to better understand its function in periodontal diseases.

The pathway analysis results align with previous studies, supporting the involvement of specific metabolic pathways in periodontitis and obesity, and reinforcing the value of unstimulated saliva samples in identifying metabolites associated with these conditions.

Answer: We appreciate the reviewer’s comment and compliment, recognizing the quality of our discussion section. Thank you so much.

Reviewer 2 Report

1.       What is the cutoff like fold change for metabolites that are increased/decreased between two groups? Please specify in the method section.

2.       What do “A” and “B” mean in Table 1?

3.       In Figure 1A, what does the “red star” mean in each column? And 1B, what does “CS (ppm)”? Please add the description in figure legend to make the figures clear.

Author Response

July 07th, 2023

We would like to thank the editors and reviewers for the caution and effort at revising our manuscript entitled “Comparison of the metabolic profile between unstimulated and stimulated saliva samples from pregnant women with/without obesity and periodontitis”. The reviewers’ positive opinions regarding the merit of the present study were enriching so that we could adapt the manuscript in order to make it suitable for publication in this respectful journal. In addition, the authors point out that the quality of the review performed by the reviewers reflects the knowledge and mastery of these professionals in this research field, which contributed to improve our manuscript.

   We hope to have answered all the questions raised by the editor and reviewers, but we are available for any other questions.

  1. What is the cutoff like fold change for metabolites that are increased/decreased between two groups? Please specify in the method section.

Answer: As in the univariate analysis we used the t-test to compare US and SS, important features were selected by t-tests with threshold 0.05. In the multivariate analysis, we considered the configuration of the MetaboAnalyst software and selected the top 15 metabolites presented by the software in the VIP score of the PLS-DA. There are two importance measures in PLS-DA: one is variable importance in projection (VIP) and the other is the weighted sum of absolute regression coefficients (coef.). The colored boxes on the right of each figure in the manuscript (Figure 1–4) indicate the relative concentrations of the corresponding metabolite in each subgroup under study (in this case, US and SS samples). Thank you for this question, this information was better detailed in the methodology.

  1. What do “A” and “B” mean in Table 1?

Answer: We included at the bottom of table 1 that different letters indicate differences between groups. Thank you for this comment.

  1. In Figure 1A, what does the “red star” mean in each column? And 1B, what does “CS (ppm)”? Please add the description in figure legend to make the figures clear.

Answer: Thank you once again for your comment. Red stars indicate p < 0.05 in the correlation and CS (ppm) means “chemical shift” of the 1H-NMR, represented by parts per million (ppm). We included this information at the bottom of each figure (Figures 1–4). Thank you.

Round 2

Reviewer 1 Report

The authors have addressed to my comments